# Insulin-like Growth Factor-1 (IGF-1) Related Drugs in Pain Management

**DOI:** 10.3390/ph16050760

**Published:** 2023-05-18

**Authors:** Seokhyun Jin, Jianguo Cheng

**Affiliations:** 1Department of Pain Management, Neurological Institute, Cleveland Clinic, Cleveland, OH 44106, USA; 2Department of Neurosciences, Lerner Research Institute, Cleveland Clinic, Cleveland, OH 44106, USA; 3Departments of Pain Management and Neurosciences, Cleveland Clinic, 9500 Euclid Avenue/C25, Cleveland, OH 44195, USA

**Keywords:** IGF-1, IGF-1R inhibitor, pain management

## Abstract

*Objective*. The aim of this review is to explore the role of IGF-1 and IGF-1R inhibitors in pain-related conditions and assess the effectiveness of IGF-1-related drugs in pain management. Specifically, this paper investigates the potential involvement of IGF-1 in nociception, nerve regeneration, and the development of neuropathic pain. *Methods.* We conducted a search of the PUBMED/MEDLINE database, Scopus, and the Cochrane Library for all reports published in English on IGF-1 in pain management from origination through November 2022. The resulting 545 articles were screened, and 18 articles were found to be relevant after reading abstracts. After further examination of the full text of these articles, ten were included in the analysis and discussion. The levels of clinical evidence and implications for recommendations of all the included human studies were graded. *Results.* The search yielded 545 articles, of which 316 articles were deemed irrelevant by reading the titles. There were 18 articles deemed relevant after reading abstracts, of which 8 of the reports were excluded due to lack of IGF-1-related drug treatment after reviewing the full text of the articles. All ten articles were retrieved for analysis and discussion. We found that IGF-1 may have several positive effects on pain management, including promoting the resolution of hyperalgesia, preventing chemotherapy-induced neuropathy, reversing neuronal hyperactivity, and elevating the nociceptive threshold. On the other hand, IGF-1R inhibitors may alleviate pain in mice with injury of the sciatic nerve, bone cancer pain, and endometriosis-induced hyperalgesia. While one study showed marked improvement in thyroid-associated ophthalmopathy in humans treated with IGF-1R inhibitor, two other studies did not find any benefits from IGF-1 treatment. *Conclusions.* This review highlights the potential of IGF-1 and IGF-1R inhibitors in pain management, but further research is needed to fully understand their efficacy and potential side effects.

## 1. Introduction

Chronic pain affects at least 116 million U.S. adults and it costs $560 to 635 billion annually in direct medical treatment costs and lost productivity, which is a major public health issue [1]. Disturbance of the GH/IGF-1/ghrenlin paracrine axis has been linked to conditions with chronic pain [2,3]. Treatment with GH for fibromyalgia and chronic lower back pain syndromes has shown effectiveness in human studies [4,5,6], as we have recently reviewed [7,8]. However, there has only been a limited number of studies of IGF-1-related drugs on chronic pain. 

Recently, there has been a growing interest in the role of IGF-1 in the peripheral and central nervous systems. IGF-1 promotes the growth, survival, and differentiation of neuronal cells both in vivo and in vitro [9,10]. It may also regulate several nerve cell functions [11]. IGF-1 in physiological concentrations promotes the survival of animal sympathetic neurons [12] Additionally, multiple animal experiments showed a significant increase in IGF-1 immunoreactivity and/or mRNA in peripheral nerves damaged by vibration [13], freeze injury [14], and crush injury [15]. This increase is in proportion to the degree of injury as well as temporally related to the noxious process making it possible that IGF-1 may play a role in the regenerative process [16]. IGF-2 antisera, which also cross-reacts with IGF-1 infusions, inhibited both motor and sensory nerve regeneration after axotomy suggesting that endogenous IGFs may play an important physiological role in nerve regeneration [17]. IGF-1 improves peripheral nerve function in a diabetic animal model [18]. Coadministration of IGF-1 has also shown efficacy in neuropathy induced by the chemotherapy agents (vincristine and taxol) in mice [19]. Both diabetic neuropathy and chemotherapy-induced neuropathy are among the most common causes of neuropathic pain [20,21].

IGF-1 is an anabolic neurotrophin serving a vital function in brain development, maturation, and neuroplasticity [22]. The IGF-1R expression is high in the developing brain and remains widely expressed in the adult nervous system [23]. However, the function of IGF-1/IGF-1R signaling in pain modulation is still controversial. Intrathecal administration of IGF-1 induced a central antinociceptive effect in normal rats [24]. However, peripheral IGF-1 contributes to pain behaviors caused by tissue injury [25] and orofacial neuropathic pain [26]. It is unclear whether and how IGF-1 participates in the pathogenesis of neuropathic pain. The aim of this review is to systematically review published data, which includes animal and human studies, on the outcomes of IGF-1-related drugs on chronic pain conditions. 

## 2. Methods

We conducted searches on the literature from PUBMED, Scopus, and the Cochran Library for all reports published in English on IGF-1 in pain management from origination through November of 2022. Search terms included “insulin-like growth factor-1” and “pain”. The following is an example of the query that was performed for the PUBMED/MEDLINE database: (“insulin-like growth factor i” [MeSH Terms] OR “insulin-like growth factor i” [All Fields] OR “insulin-like growth factor 1” [All Fields]) AND (“pain” [MeSH Terms] OR “pain” [All Fields]). Both human and animal studies were included in the search results. A further manual search was performed to exclude any duplicate records and irrelevant articles by screening the titles and the abstracts. The abstracts were subsequently reviewed, and the full-text publications were retrieved when it was necessary. Additional literature was further identified after reading the full text of the relevant articles. The studies without an actual intervention with IGF-1-related drugs were excluded after reading the full text of the articles.

The levels of clinical evidence and implications for recommendations of all the included human studies were graded based on “Grading Strength of Recommendations and Quality of Evidence in Clinical Guidelines” [27] (Appendix A). This method of the level of evidence classification takes into consideration the type of the studies, the quality of the studies, and the number of studies. All the authors independently performed the search and extracted data from the articles. Any disagreements were resolved by discussion. Data were extracted from each relevant publication on study design including study setting, data source, study period, dose of the medication, route, and side effects.

## 3. Results

The search strategy yielded 545 articles, of which 316 articles were deemed irrelevant by reading the titles. There were 18 articles left after reading abstracts, 8 of which were excluded due to lack of IGF-1 related drug treatment after reviewing the full text of the articles (Figure 1). Through the systematic review of ten articles published between 1997 and 2021, we identified seven observational studies in animals and three randomized controlled trials in humans. 

The listed studies were divided into three groups. The first group included studies that investigated outcomes from IGF-1-related drugs on neuropathic pain in animals. The second group included studies that investigated outcomes from IGF-1-related drugs on non-neuropathic pain in animals. The third group included studies that explored the efficacy of IGF-1-related drugs on any pain-related disease in humans. 

### 3.1. IGF-1 on Neuropathic Pain

Neuropathic pain is a chronic condition induced by damages disrupting the central or peripheral somatosensory nervous system [28]. Chronic neuropathic pain is usually intractable to the existing pain treatments, making it a critical need to develop new therapies [29]. IGF-1R expression is high in the developing brain and remains widely expressed in the adult nervous system [23].

IGF-1 signaling may play a significant role in the pathogenesis of neuropathic pain. Chen et al. [30] tested the hypothesis that IGF-1 signaling might be involved in the pathogenesis of neuropathic pain by regulating mTOR mediated autophagy. They injected nvp-aew541 (IGF-1R inhibitor) and anti-IGF-1 neutralizing antibodies into mice with CCI (chronic constriction injury of the sciatic nerve) to test the function of the IGF-1/IGF-1R pathway in mediating neuropathic pain. Both nvp-aew541 and anti-IGF-1 neutralizing antibodies reduced mechanical allodynia (assessed by paw mechanical withdrawal thresholds (PWT) to von Frey filament stimulation) and thermal hyperalgesia (assessed by paw withdrawal latency (PWL) to thermal stimulation). In addition, both agents markedly suppressed the spinal p-IGF-1R expression in CCI mice on day 14, significantly counteracted the increase of p-mTOR and p-S6K, and attenuated the CCI-induced decrease in p62, beclin-1, and the apidated LC311 to nonlipidated LC31 ratio. Furthermore, both agents decreased the CCI-induced production of IL-1beta, TNF alpha, and IL 6. These results suggest that IGF-1R antagonism or IGF-1 neutralization alleviated the pain-related behaviors, relieved the mTOR-induced suppression of autophagy, and mitigated neuroinflammation induced by CCI in mice.

However, there is evidence that IGF-1 signaling may contribute to the resolution of acute pain caused by tissue injury, thereby mitigating acute-to-chronic pain transition. Takemura et al. [31] investigated the expression of G-protein coupled receptor kinase 2 (GRK2) after plantar incision in rats. Animals in the IGF-1 group received an ipsilateral intraplantar injection of IGF-1 (1 µg/50 µL/rat) and the contralateral IGF-1 group received an IGF1 injection in the contralateral paw to investigate the effect of IGF-1 on GRK2 expression in the DRG. Rats in the incision with IGF1R inhibitor group subsequently received an injection of the IGF-1R inhibitor, picropodophyllin (dissolved in 50 µL of 10% DMSO) in the ipsilateral plantar area 1, 3 and 5 days after incision. Two doses (250 µg/kg and 500 µg/kg) of the IGF1R inhibitor were tested to assess the effect of IGF-1 on nociceptive control during the recovery period of postoperative pain. Plantar incision induced an increase in GRK2 in the DRG at 7 days, but not at 1 day post-incision. Acute hyperalgesia, after the plantar incision, disappeared by 7 days post-incision. Intraperitoneal injection of the GRK2 inhibitor reinstated mechanical hyperalgesia in plantar incision rats but not in naive rats. After the incision, IGF-1 expression increased in the paw, but not in the DRG. Intraplantar injection of IGF-1 increased GRK2 expression in the ipsilateral DRG. IGF-1R inhibitor administration prevented both the induction of GRK2 and the resolution of hyperalgesia after the plantar incision. These findings demonstrate that the induction of GRK2 expression driven by tissue IGF-1 has potent analgesic effects and produces a resolution of hyperalgesia after tissue injury. Dysregulation of IGF-1-GRK2 signaling may potentially lead to failure of resolution of acute pain, and hence, the development of chronic pain after surgery.

IGF-1 signaling may also mitigate chemotherapy-induced neuropathy (CINP). Contreras et al. [32] demonstrated that administration of IGF-1 attenuates both motor and sensory manifestations of vincristine-induced neuropathy in a dose-related fashion. CINP was induced by intraperitoneal vincristine injections twice a week for 10 weeks (1.7 mg/kg) and 0.3 or 1 mg/ kg of IGF-1 during the period of study. IGF-1 at a dose of 1 mg/kg, but not 0.3 mg/kg, prevented the adverse effect of vincristine on gait and stride lengths. Additionally, IGF-1 0.3 or 1 mg/kg prevented the increase in hot-plate latencies observed in vincristine-treated mice. The beneficial effects of IGF-1 appeared to be dose-dependent, as mice treated with a higher dose of IGF-1 1 mg/kg had hot-plate latencies that were significantly improved than those treated with a lower dose of IGF-1 0.3 mg/kg. Animals treated with IGF-1 0.3 mg/kg showed a significant improvement in nerve fiber, which was further improved in animals treated with 1 mg/kg of IGF-1. The study demonstrates that the coadministration of IGF-1 with vincristine prevented behavioral and histopathological manifestations of both sensory and motor dysfunction in a dose-dependent fashion. These findings support potential clinical trials of IGF-1 in the treatment of vincristine neurotoxicity and possibly other mixed neuropathies.

IGF-1 may also mitigate painful diabetic neuropathy (PDN). Morgado et al. [33] evaluated if treatment with IGF-1 affects the behavioral signs of PDN, neuronal hyperactivity at the spinal cord and at the periaqueductal gray matter of the midbrain (PAG), and serotoninergic and noradrenergic brainstem systems. The STZ (streptozocin)-diabetic rats were injected subcutaneously with 2.5 mg/kg of recombinant human IGF-1 or saline. Treatments with IGF-1 reversed *Fos* expression to the control levels at the spinal dorsal horn and the ventrolateral part of PAG (VLPAG) in STZ-diabetic rats, but had no effects in non-diabetic control animals. Treatments with IGF-1 prevented the increased levels of serotonin at the spinal cord and the rostroventromedial medulla (RVM) in STZ-diabetic rats, but had no effects on control animals. IGF-1 treatment had no effects on paw withdrawal threshold (PWTs) in control animals nor in animals treated with formalin injections. The study demonstrated that treatment with IGF-1 prevented the behavioral signs of PDN and reversed the neuronal hyperactivity and neurochemical changes in the spinal cord and the brainstem.

Bitar et al. [34] examined the cellular mediators for the elevation of the nociceptive threshold induced by IGF-1 in STZ-treated rats. Drugs containing genistein (tyrosine kinase inhibitor) were administered intrathecally 15 min prior to treatment of 0.2, 0.4, 0.6, or 1 µg of intrathecal IGF-1. Intrathecal IGF-1 administered at a dose of 0.6 µg significantly elevated the nociceptive threshold over saline-treated animals by about 35%. This effect was blocked by genistein at doses, that in themselves, did not have any significant effect on pain threshold. mRNA transcripts for IGF-1 and its receptor in the spinal cord were reduced in STZ-diabetic rats, which showed a reduced nociceptive threshold. Additionally, they showed the antinociceptive effect of IGF-1 may also be mediated by a similar type of receptor sensitive to genistein. 

### 3.2. IGF-1 in Other Pain Conditions

IGF-1 may contribute to pain pathogenesis in cancer by upregulating the expression and function of TRPV1. Li et al. [35] found that IGF-1 secretion was increased in local skeletal tissue following cancer cell invasion into bone marrow, subsequent bone destruction, and reconstruction. IGF-1 was dissolved and diluted in sterile PBS (phosphate-buffered saline) solution to the desired concentration (3, 30, and 100 ng/mL). IGF-1R inhibitor picropodophyllotoxin (PPP) at 20 mg/kg/12 h was injected intraperitoneally for 3 consecutive days, from days 15 to 17, after MRMT-1 (rat mammary gland carcinoma cells) live cell inoculation into tibia bone marrow. The increase of IGF-1 led to increased expression and function of TRPV1 (transient receptor potential vanilloid subfamily member 1) in the nerve fiber endings innervating the bone membrane. The effect of IGF-1 was further tested on primary cultured DRG neurons. The total TRPV1 expression was increased significantly at 48 and 72 h in the presence of IGF-1 at 30 and 100 ng/mL, as compared with PBS control. Incubation with IGF-1 at 30 ng/mL for 24 and 72 h significantly increased capsaicin-induced currents in DRG neurons, suggesting that IGF-1-upregulated TRPV1 is functional. Furthermore, the IGF-1R inhibitor significantly alleviated pain behaviors suggesting that IGF-1R inhibition could reverse cancer pain in rats. These results suggest that an enhanced TRPV1 function via IGF-1 upregulation in metastasized bone cancer pain and IGF-1 upregulation on TRPV1 through IGF-1R contributes to cancer pain. 

Macrophage-derived IGF-1 may play a role in endometriosis-associated pain. Forster et al. [36] studied the mechanistic role of macrophages in producing pain associated with endometriosis, and showed that macrophage depletion in a mouse model of endometriosis can reverse abnormal changes in pain behavior. Linsitinib, which is an IGF1R inhibitor (40 mg/kg) with modest activity of the insulin receptor, was administered by oral gavage every 24 h. They identified that disease-modified macrophages exhibit increased expression of IGF-1 in an in vitro model of endometriosis-associated macrophages, and confirmed expression by lesion-resident macrophages in mice and women. Concentrations of IGF-1 were elevated in peritoneal fluid from women with endometriosis and positively correlated with their pain scores. Mechanistically, macrophage-derived IGF-1 promotes sprouting neurogenesis and nerve sensitization in vitro and the IGF-1 receptor inhibitor Linsitinib reverses the pain behavior observed in mice with endometriosis. These data support the role of macrophage-derived IGF-1 as a key neurotrophic and sensitizing factor in endometriosis. Therapies that modify the macrophage phenotype may be attractive therapeutic options for the treatment of women with endometriosis-associated pain. 

### 3.3. IGF-1 on Pain in Humans

Three clinical applications of IGF-1-related drugs in pain conditions are tabulated in Appendix A. Olesen et al. [37] hypothesized that injection of IGF-1 into the tendons of persons with patellar tendinopathy (PT), in combination with heavy slow resistance (HSR) training, would enhance patellar tendon collagen synthesis and provide a faster recovery than HSR training alone. In a double-blind randomized controlled trial, 40 patients with unilateral PT diagnosed by an experienced sports physician were randomized to intratendinous injection of either IGF-1 or isotonic saline over 3 weeks, accompanied by 12 weeks of HSR training in both groups. Three intratendinous injections of either IGF-1 (0.1 mL, 10mg/mL) or isotonic saline injections (0.1 mL isotonic saline) were conducted at weeks 0, 1, and 2. The HSR training was initiated the same day as the first injection and continued for 12 weeks. They found a significant decrease in Doppler activity score only in the IGF-1 group after 12 weeks and a significant decline in tendon thickness only in the placebo group. No significant difference was observed between the two groups in the VISA-P (Victoria Institute of Sport Assessment–Patella) score, nor in VAS scores, after 12 weeks. The study did not demonstrate that adding IGF-1 to a strength training program had any positive clinical or biochemical effects on PT. They concluded that intratendinous IGF-1 injections with HSR training were safe and well tolerated but did not show any additive improvement in tendon healing compared with resistance training alone.

Windebank et al. [38] conducted a study to investigate the effects of IGF-1 on patients with painful distal and symmetric neuropathy. A total of 40 patients over 18 years old, with distal neuropathic pain or hyperalgesia syndrome, were equally randomized to either IGF-1 or placebo treatment for 6 months. The results showed no significant difference in pain relief or other secondary endpoints between the two groups, and no serious side effects were observed in the IGF-1 group. Therefore, the study suggests that IGF-1 can be safely given to patients but may not be beneficial in treating painful small fiber predominant neuropathy.

Smith et al. [39] studied the clinical outcomes and safety of teprotumumab, an IGF-1R inhibitory monoclonal antibody, as a treatment for thyroid-associated ophthalmopathy in 88 randomized patients aged 18–75. Patients received eight intravenous infusions of teprotumumab, starting with a dose of 10 mg/kg, followed by 20 mg/kg for the remaining seven infusions, over a 24-week period. Results showed marked improvement in the primary outcome measure, time to the first response, and onset of the response in the teprotumumab group, as well as improved Clinical Activity Score, proptosis, and GO-QOL visual functioning score. Serious adverse events were reported in 5 of 43 patients in the teprotumumab group, including two cases of diarrhea and mental confusion which were categorized as “possibly related” to the drug by the investigators. Thus, the study concluded that teprotumumab can safely provide clinical benefit in patients with active, moderate-to-severe, thyroid-associated ophthalmopathy.

The summary of the above ten studies regarding IGF-1 related drugs in pain conditions (Appendix A) and IGF-1R inhibitors in pain conditions (Appendix A) are tabulated. 

## 4. Discussion

We found that four studies have shown favorable outcomes of recombinant IGF-1 on chronic pain in animals. Takemura et al. showed that IGF-1 promotes acute resolution of hyperalgesia [31], Contreras et al. suggested IGF-1 prevents vincristine-induced neuropathy [32], Morgado et al. found that IGF-1 prevented behavioral signs of PDN and reversed neuronal hyperactivity [33], and Bitar et al. showed IGF-1 elevates nociceptive threshold and its antinociceptive effects are mediated by tyrosine kinase inhibitors [34]. We found that three animal studies and one human study demonstrated favorable outcomes with IGF-1R inhibitors. Chen et al. suggested that IGF-1R antagonism alleviates pain-related behaviors and neuroinflammation [30], Li et al. found that IGF-1R inhibitor alleviated pain behaviors [35], Forster et al. showed that IGF-1R inhibition attenuates hyperalgesia [36], and Smith et al. found that teprotumumab reduced proptosis in the Clinical Activity Score and improved quality of life in patients with thyroid-associated ophthalmopathy [39].

Several studies have demonstrated improvements in behavioral signs of diabetic neuropathy after IGF-1 treatment. One study showed that continuous subcutaneous infusion of IGF-1 halted the advancement of hyperalgesia in STZ-diabetic rats, while another suggested that IGF-1 has a key role in maintaining the health and balance of the nervous system. These studies suggest that IGF-1 treatment may be beneficial for chronic pain.

Two studies involving human subjects showed no benefit from IGF-1 treatment for chronic pain. Olesen et al. found that intratendinous IGF-1 injections with HSR training were safe and well-tolerated, but did not improve tendon healing compared to resistance training alone. Windebank et al. found that IGF-1 can be safely administered, but there was no evidence that it was beneficial for patients with painful small fiber predominant neuropathy. On the other hand, one study showed that IGF-1 has a dual role in relation to traumatic brain injury. Initially, IGF-1 is neuroprotective, but long-term exposure leads to persistent neuronal hyperexcitability. It is thought that spinal IGF-1 increases in response to initial nerve injury as a protective factor, but the excessive release of IGF-1 can worsen neuropathic pain by causing neuronal hyperexcitability. These findings suggest that IGF-1 treatment may not be beneficial for pain-related diseases.

It Is known that IGF-1R activation initiates the phosphoinositide 3-kinase (PI3K)-Akt pathway, which controls a variety of cellular functions, such as protein synthesis and apoptosis. Akt is an effector of this pathway and has been linked to nerve-injury-induced neuropathic pain. IGF-1R activation also activates mTOR signaling, which is necessary for the development of painful diabetic neuropathy. IGF-1, found in peritoneal fluid, is thought to contribute to endometriosis by stimulating the growth and preventing apoptosis of endometrial-like cells. IGF-1 can cause pain hypersensitivity and binds to IGF-1R to activate a variety of intracellular signaling pathways. IGF-1R is expressed in small, medium, and large DRG neurons, and in chronic inflammatory and tissue injury models, IGF-1 signaling increases thermal and mechanical hyperalgesia. These studies suggest that IGF-1R inhibition may reduce pain through the above-mentioned mechanisms.

IGF-I is considered a potential therapeutic agent for its powerful neurotrophic effects [40,41], as evidenced by successful results in animal models of neuropathy using similar doses [19,32]. Specifically, dorsal root ganglions have been found to express IGF-1 receptors and respond to their neurotrophic effects [42]. However, while there is no evidence of IGF-1 treatment being beneficial in humans, Windebank et al. [38] suggest that this lack of efficacy may be due to different underlying biological processes, insufficient treatment length, or less sensitive assessment instruments. They also propose that chronic peripheral denervation-induced central remodeling may hinder symptomatic improvement despite peripheral regeneration.

Three human studies have shown that IGF-1 treatment or an IGF-1R inhibitor can be administered safely to patients. The positive safety profile of IGF-1 treatment has also been established in other studies involving patients with amyotrophic lateral sclerosis (ALS) [43,44]. Moreover, the favorable safety profile of teprotumumab, an IGF-1R inhibitor, in patients with ophthalmopathy is consistent with that observed in previous oncology studies [45,46] as well as a study on other anti-IGF-1R antibodies [47].

### Research Gaps and Prospects

We found mixed outcomes of recombinant IGF-1 and IGF-1R inhibitors on chronic pain in animals and humans. Recombinant IGF-1 promoted acute resolution of hyperalgesia [31], prevented vincristine-induced neuropathy [32], and behavioral signs of PDN and reverse neuronal hyperactivity [33], and elevated nociceptive threshold and its antinociceptive effects [34]. However, recombinant IGF-1 failed to show any benefit in humans treating the pain associated with patellofemoral tendinopathy and painful small fiber neuropathy [37,38]. On the other hand, IGF1R inhibitor alleviated pain behaviors [35], attenuated hyperalgesia [36], and teprotumumab (IGF1R inhibitor) reduced proptosis, the Clinical Activity Score and improved quality of life in patients with thyroid associate ophthalmopathy [39]. There is only one human study that showed the benefit of IGF1R inhibitors in chronic pain conditions and more studies need to be performed to investigate the safety and efficacy of IGF1R inhibitors in chronic pain conditions in humans in the future. Additionally, we were not able to find an explanation of why both recombinant IGF-1 and IGF1R inhibitors alleviate pain conditions in animals even though the pharmacologic action of both drugs is thought to be contrary. Takemura et al. [31] demonstrated IGF-1R inhibitors prevented both the induction of GRK2 and the resolution of hyperalgesia which is different from the other studies that showed the favorable outcome of IGF1R inhibitors in pain conditions. Based on these contradictory results, more research is needed to determine the safety and efficacy of IGF1-related drugs.

## 5. Concluding Remarks

Studies show that IGF-1 may promote the resolution of hyperalgesia, prevent chemotherapy-induced neuropathy and behavioral signs of PDN, reverse neuronal hyperactivity, and elevate the nociceptive threshold. However, there are studies that suggest IGF-1R antagonism may alleviate pain-related behaviors in mice with injury of the sciatic nerve, pain behaviors from bone cancer, and hyperalgesia from endometriosis. In addition, IGF-1R inhibitor showed marked improvement in thyroid-associated ophthalmopathy in humans even though two other human studies did not demonstrate benefits from IGF-1 treatment. IGF-1R inhibition has been widely investigated for various cancer treatments with numerous clinical trials [48]. On the contrary, the knowledge of IGF-1-related drugs for pain-related diseases is slowly evolving, and its efficacy is still in question because of a lack of advanced evidence. With contradictory results from these studies, no definitive recommendations can be made at this point and more research is needed to determine the efficacy and safety of intervening in this signaling pathway and to guide future clinical applications in pain medicine.

## Figures and Tables

**Figure 1 pharmaceuticals-16-00760-f001:**
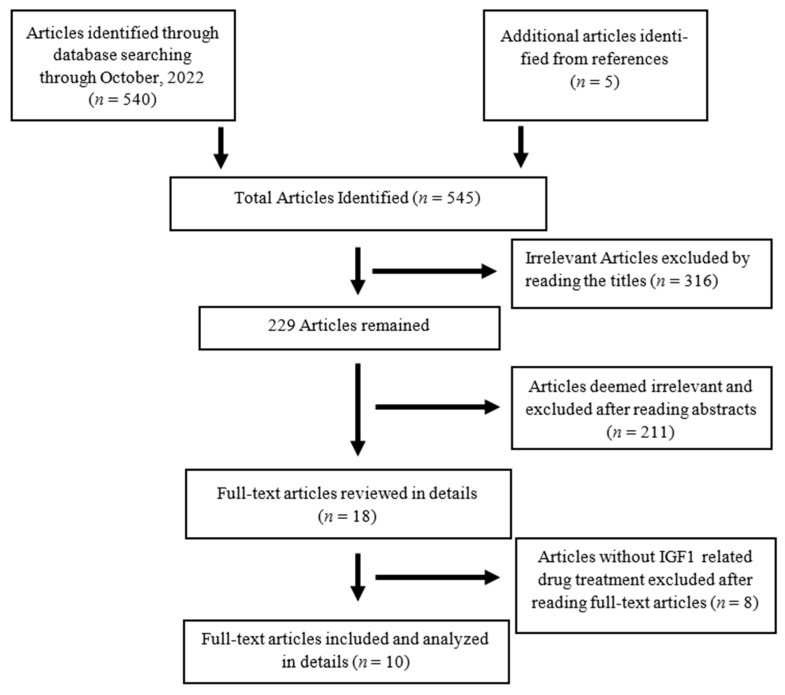
Flow diagram of literature search and process of inclusion/exclusion of articles.

## Data Availability

Data sharing not applicable.

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
