# Peer review of "Insulin-like Growth Factor-1 (IGF-1) Related Drugs in Pain Management"

_pharmaceuticals, 2023, doi:10.3390/ph16050760_

Round 1
Reviewer 1 Report
The authors indicated on the Methods section that "A systematic review of original articles was conducted according to the Preferred Reporting Items for Systematic review and Meta-Analyses Protocols (PRISMA-P)." However, apart from figure 1, I was not able to find any of the other important items for PRISMA-P reporting, such as
-Study risk of bias assessment
-Synthesis methods
-Certainty assessment
-Registration and protocol
and so on. I suggest the authors to check the paper "The PRISMA 2020 statement: an updated guideline for reporting systematic reviews" https://doi.org/10.1136/bmj.n71. If the authors want to follow those guidelines, there is a useful website with various indication for PRISMA-P guidelines.
Moreover, at the end of the selection procedure, the authors found just 3 article worth of reviewieng and they choose to not make a meta-analysis of the data; thus not making any conclusive assessment and lowering the impact of the whole study.
In conclusion, the authors should decide if they want to make a systematic review and meta-analysis of the data, rewriting fully the paper or make a narrative review and then submitting to another journal
Author Response
Thank you to the reviewers and editors for insightful comments and helpful suggestions, which have significantly helped improve the quality of the review
The authors indicated on the Methods section that "A systematic review of original articles was conducted according to the Preferred Reporting Items for Systematic review and Meta-Analyses Protocols (PRISMA-P)." However, apart from figure 1, I was not able to find any of the other important items for PRISMA-P reporting, such as …
Response: We systematically conducted a search; however, our study did not meet the criteria for a systematic review according to PRISMA-P due to the limited number of research articles available in the literature (ie; only 3 RCTs – human studies). Therefore, we submit this paper as a narrative review article. We removed the first line in the Method section.

Reviewer 2 Report
1. Authors should describe the results of 10 studies that include in this systematic review in table. Authors can describe the results of study, the methods of every study in table.
2. Authors should declare the novelty of systematic? Why the systematic review is needed to be done?
Author Response
Thank you to the reviewers and editors for insightful comments and helpful suggestions, which have significantly helped improve the quality of the review.
Authors should describe the results of 10 studies that include in this systematic review in table. Authors can describe the results of study, the methods of every study in table.
Response: Excellent suggestion. Table 3 is added.
Table 3. Summary of 10 studies regarding IGF-1-related drugs in pain conditions
|
Reference |
Methods |
Results |
Conclusion |
|
Chen et al31 |
Intrathecal injection of IGF-1R inhibitor and anti-IGF1 neutralizing antibodies to mice with CCI |
Both IGF-1R inhibitor and anti-IGF-1 neutralizing antibodies reduced mechanical allodynia and thermal hyperalgesia in mice with CCI. |
IGF-1R antagonism and IGF-1 neutralization alleviated the pain-related behaviors, relieved the mTOR-induced suppression of autophagy, and mitigated neuroinflammation induced by CCI in mice. |
|
Takemura et al32 |
Intra-plantar injection of IGF-1 and IGF-1R inhibitors to rats after plantar incision |
IGF-1 increased GRK2 expression in the ipsilateral DRG. IGF-1R inhibitor prevented both the induction of GRK2 and the resolution of hyperalgesia. |
IGF-1R inhibition leads to failure of spontaneous resolution of hyperalgesia after tissue injury. Dysregulation of IGF-1-GRK2 signaling might be one of the major pathological conditions leading to the transition from acute to chronic pain after surgery. |
|
Contreras et al33 |
Subcutaneous recombinant IGF-1 injection to mice with chemotherapy-induced neuropathy |
IGF-1 prevented the increase in hot-plate latencies and showed significant improvement in nerve fibers in vincristine-treated mice. |
Coadministration of IGF-1 with vincristine prevented behavioral and histopathological manifestations of both sensory and motor dysfunction in a dose-dependent fashion. |
|
Morgado et al34 |
Subcutaneous recombinant IGF-1 injection in STZ-diabetic rats |
IGF-1 reversed Fos expression to the control levels at the spinal dorsal horn and the VLPAG and prevented the increased levels of serotonin at the spinal cord and the RVM in STZ diabetic rats. |
IGF-1 prevented the behavioral signs of PDN and reversed the neuronal hyperactivity and neurochemical changes at the spinal cord and at the brainstem. |
|
Bitar et al35 |
Intrathecal IGF-1 injection in STZ-diabetic rats |
IGF-1 elevated the nociceptive threshold over saline-treated animals by about 35%. mRNA transcripts for IGF-1 and its receptor in the spinal cord were reduced in STZ-diabetic rats, which showed a reduced nociceptive threshold. |
The attenuation in the ability of IGF-1 to elevate the nociceptive threshold may be a consequence of reduced gene expression of the IGF-1 receptor within the spinal cord. |
|
Li et al36 |
Intraperitoneal IGF-1 and IGF-1R inhibitor injection in rats with MRMT1 bone cancer pain |
IGF-1 increased the expression and function of TRPV1 and significantly increased capsaicin-induced currents in DRG neurons. IGF-1R inhibitor significantly alleviated pain behaviors. |
An enhanced TRPV1 function via IGF-1 upregulation in metastasized bone cancer pain and IGF-1 upregulation on TRPV1 through IGF-1R contributes to cancer pain. |
|
Forster et al37 |
Macrophage IGF-1 receptor inhibitor (Linsitinib) was injected through oral gavage in mice with endometriosis. |
IGF-1 receptor inhibitor reverses the pain behavior observed in mice with endometriosis |
Therapies that modify macrophage phenotype may be attractive therapeutic options for the treatment of women with endometriosis-associated pain |
|
Olesen et al38 |
Intratendinous IGF-1 injection in patients with patellar tendinopathy |
No significant difference in VAS score, VISA-P score, or biochemical effect compared to the control group |
Intratendinous IGF-1 injections with HSR training were safe and well tolerated but did not show any additive improvement in tendon healing compared with resistance training alone |
|
Windebank et al39 |
Subcutaneous IGF-1 injection in patients with painful distal and symmetric neuropathy |
No significant difference in the analog pain scale. CASE vibratory threshold favored the placebo group while WBPI walking ability favored the IGF-1 treatment group |
IGF-1 can be safely given to patients but may not be beneficial in treating painful small fiber predominant neuropathy
|
|
Smith et al40 |
Intravenous IGF-1R inhibitory monoclonal antibody (teprotumumab) injection in patients with thyroid-associated ophthalmopathy |
Marked improvement in the primary outcome measure, time to the first response and onset of the response, Clinical Activity Score, proptosis, and GO-QOL visual functioning score in the teprotumumab group |
Teprotumumab can safely provide clinical benefit in patients with active, moderate to severe associated ophthalmopathy |
Abbreviations: IGF-1; insulin-like growth factor-1, IGF-1R; insulin-like growth factor-1 receptor, CCI; chronic constriction injury, mTOR; mechanistic target of rapamycin, GRK2; G-protein coupled receptor kinase-2, DRG; dorsal root ganglion, STZ; streptozocin, VLPAG; ventrolateral periaqueductal gray, RVM; rostroventromedial medulla, PDN; peripheral diabetic neuropathy, MRMT-1; rat mammary gland carcinoma cells-1 , TRPV-1; transient receptor potential vanilloid subfamily member 1, VAS; visual analog scale, VISA-P; Victorian institute of sport assessment – patella, HSR; heavy slow resistance, CASE; computer assisted sensory examination, WBPI; Wisconsin brief pain inventory, GO-QOL; Graves ophthalmopathy specific quality of life questionnaire

Reviewer 3 Report
This mini-review is informative in some extent; however, needs more information to be included for the readers.
A more descriptive paragraphs explaining the 'Research gaps and prospects' (with the same heading) of IGF-1 in the pain management needs to be included, which would help the researchers to focus these questions.
A table mentioning the IGF-1R inhibitors/drugs in the management of pain associated diseases may be comprehensively required to strengthen this manuscript.
The result and discussion may be comprehended to sub sections.
Line 40 and 321: behavior is odd in the sentences
Line 77: synthesize published data?
Author Response
Thank you to the reviewers and editors for insightful comments and helpful suggestions, which have significantly helped improve the quality of the review.
Research gaps and prospects' (with the same heading) of IGF-1 in the pain management needs to be included, which would help the researchers to focus these questions.
Response: This is an excellent suggestion. The following paragraph is added.
Research gaps and prospects We found mixed outcomes of recombinant IGF-1 and IGF-1R inhibitors on chronic pain in animals and humans. Recombinant IGF-1 promoted acute resolution of hyperalgesia (32), prevented vincristine-induced neuropathy (33), and behavioral signs of PDN and reverse neuronal hyperactivity (34), and elevated nociceptive threshold and its antinociceptive effects (35). However, recombinant IGF-1 failed to show any benefit in humans treating the pain associated with patellofemoral tendinopathy and painful small fiber neuropathy. (38,39) On the other hand, IGF1R inhibitor alleviated pain behaviors (36), attenuated hyperalgesia (37), and teprotumumab (IGF1R inhibitor) reduced proptosis, the Clinical Activity Score and improved quality of life in patients with thyroid associate ophthalmopathy (40). There is only one human study that showed the benefit of IGF1R inhibitors in chronic pain conditions and more studies need to be performed to investigate the safety and efficacy of IGF1R inhibitors in chronic pain conditions in humans in the future. Also, we were not able to find an explanation of why both recombinant IGF-1 and IGF1R inhibitors alleviate pain conditions in animals even though the pharmacologic action of both drugs is thought to be contrary. Takemura et al (28) demonstrated IGF-1R inhibitors prevented both the induction of GRK2 and the resolution of hyperalgesia which is different from the other studies that showed the favorable outcome of IGF1R inhibitors in pain conditions. Based on these contradictory results, more research is needed to determine the safety and efficacy of IGF1-related drugs.
A table mentioning the IGF-1R inhibitors/drugs in the management of pain associated diseases may be comprehensively required to strengthen this manuscript
Response: We agree. Table 4 is added.
Table 4. Summary of IGF-1R Inhibitors in Pain Conditions
|
Pain Conditions |
Reference |
Medication |
Result |
Conclusion |
|
Chronic Constriction Injury (animal) |
Chen et al31 |
IGF1R inhibitor (nvp-aew541) Anti-IGF1 neutralizing antibody |
Both IGF-1R inhibitor and anti-IGF-1 neutralizing antibodies reduced mechanical allodynia and thermal hyperalgesia in mice with CCI. |
IGF-1R antagonism and IGF-1 neutralization alleviated the pain-related behaviors, relieved the mTOR-induced suppression of autophagy, and mitigated neuroinflammation induced by CCI in mice. |
|
Metastatic bone cancer pain (animal) |
Li et al36 |
IGF1R inhibitor (PPP) |
IGF-1 increased the expression and function of TRPV1 and significantly increased capsaicin-induced currents in DRG neurons. IGF-1R inhibitor significantly alleviated pain behaviors. |
An enhanced TRPV1 function via IGF-1 upregulation in metastasized bone cancer pain and IGF-1 upregulation on TRPV1 through IGF-1R contributes to cancer pain. |
|
Endometriosis (animal) |
Forster et al37 |
IGF1R inhibitor (Linsitinib) |
IGF-1 receptor inhibitor reverses the pain behavior observed in mice with endometriosis. |
Therapies that modify macrophage phenotype may be attractive therapeutic options for the treatment of women with endometriosis-associated pain. |
|
Thyroid-associated ophthalmopathy (human) |
Smith et al40 |
IGF1R inhibitor (teprotumumab) |
Marked improvement in the primary outcome measure, time to the first response and onset of the response, Clinical Activity Score, proptosis, and GO-QOL visual functioning score in the teprotumumab group |
Teprotumumab can safely provide clinical benefit in patients with active, moderate to severe associated ophthalmopathy. |
Abbreviations: IGF-1; insulin-like growth factor-1, IGF-1R; insulin-like growth factor-1 receptor, CCI; chronic constriction injury, mTOR; mechanistic target of rapamycin, PPP; picropodophyllotoxin, TRPV-1; transient receptor potential vanilloid subfamily member 1, DRG; dorsal root ganglion, GO-QOL; Graves ophthalmopathy specific quality of life questionnaire
Line 40 and 321: behavior is odd in the sentences, Line 77: synthesize published data?
Response: The two sentences are modified for clarity.

Round 2
Reviewer 1 Report
The authors have accepted and responded to my comments. I do think the work is significantly improved.
Reviewer 3 Report
The title may be changed to Potential Insulin-Like Growth Factor-1 (IGF-1) Related Drugs in Pain Management